# Formulation of Biscuits Fortified with a Flour Obtained from Bergamot By-Products (*Citrus bergamia*, Risso)

**DOI:** 10.3390/foods11081137

**Published:** 2022-04-14

**Authors:** Valeria Laganà, Angelo Maria Giuffrè, Alessandra De Bruno, Marco Poiana

**Affiliations:** Department of AGRARIA, University Mediterranea of Reggio Calabria, 89124 Reggio Calabria, Italy; valerialagana.foodtec@gmail.com (V.L.); alessandra.debruno@unirc.it (A.D.B.); mpoiana@unirc.it (M.P.)

**Keywords:** antioxidants, bergamot by-product flour, biscuits, functional food, phenolic compounds

## Abstract

Bergamot belongs to the Rutaceae family and is a typical fruit grown principally in the province of Reggio Calabria (South Italy). Nowadays, its industrial use is mostly related to the extraction of the essential oil contained in the flavedo but also to a lesser extent the extraction of the juice (from the pulp), which is rich in antioxidants. However, the pressed pulp (known as *Pastazzo*) is either used for animal feed or is discarded. The aim of this research was to study the effect of bergamot *Pastazzo* flour in shortbread biscuits. *Pastazzo* flour partially replaced the 00 wheat flour in different percentages (2.5%, 5%, 10% and 15%). Simultaneously, a sample without the addition of *Pastazzo* flour (control) was analyzed, thus obtaining five biscuit samples. Cooking was done in a ventilated oven at 180 °C. The baking time was different for the control and the enriched samples depending on when the desired color was reached. The control took 12 min, while the enriched samples reached the desired color in 8 min. All samples were subjected to physicochemical and antioxidant characterization, as well as total polyphenols and flavonoids. The use of *Pastazzo* flour resulted in a slight increase in water activity and humidity values. pH values decreased for all the enriched samples compared to the control, but this was more relevant for the samples enriched with 10 and 15% of flour from by-products. Hardness varied from 1823 g (Control) to 2022 and 2818 g (respectively, for 2.5% and 15% bergamot *Pastazzo* flour in the recipe). Total phenol content varied from 0.14 mg GAE g^−1^ (Control) to 0.60 and 3.64 mg GAE g^−1^ (respectively, for 2.5% and 15% bergamot *Pastazzo* flour in the recipe). The obtained results demonstrated that the use of *Pastazzo* flour had a positive influence on the antioxidant content, with values which increased as more *Pastazzo* flour was added.

## 1. Introduction

Functional is a word used in the description of any food or drink containing specific nutrients or molecules that have the possibility to improve human health in addition to their recognized nutritional value. Functional foods often have antioxidant abilities, being able to neutralize free radicals and defend cells from oxidative stress [1].

Recently, the commercial importance of functional foods has grown enormously, as consumers turn to these products [2].

For this reason, the link between nutrition and health is becoming stronger and the development of new functional foods containing beneficial ingredients for health is increasing [3].

The bergamot tree is cultivated in a coastal strip of Reggio Calabria province (South Italy). The bergamot fruit has an ancient agronomic [4] and industrial tradition in Reggio Calabria. In the 2020 harvest year, 1500 hectares with bergamot plants were cultivated and 270,000 quintals of fruits were produced in the Reggio Calabria province [5].

In the past decades, the use of this fruit was limited to the essential oil extraction from the flavedo [6] aimed at the preparation of cosmetics, perfumes and sweets, and whose composition has been widely studied [7,8,9]. Nowadays, other applications are being studied to make use of bergamot by-products [10,11].

The essential oil extraction is conducted by centrifugation of the scraped flavedo. 

At this point, the scraped fruit is pressed and the juice is separated from the solid phase, leaving what is called *Pastazzo* [12], which represents about 55% of the fruit and is composed of seeds, pulps and exhausted peels. In the past, the scraped fruit was considered to be a waste product, unusable for any purpose, except for cattle feed [13]. 

Nowadays, bergamot fruit *Pastazzo* cannot be considered a waste because this by-product contains pectins, fiber and antioxidant molecules, in particular, naringin (2200 mg L^−1^), neohesperidin (15,200 mg L^−1^) and neoeriocitrin (1800 mg L^−1^) [14,15]. Additionally, many scientific works have demonstrated that the flavonoid molecules present in *Pastazzo* can contribute to preventing many metabolic diseases [16,17]. The daily intake of foods and beverages rich in phenols was found to prevent diseases such as obesity [18]; non-alcoholic fatty liver [19]; hypertension, hyperglycemia, and microbiota dysfunction [20]. Flavonoids present in the bergamot were proven to have antioxidant action and beneficial effects on human health: preventing cancer cell stems formation by targeting fatty acid oxidation and mitochondrial metabolism in breast cancer cell [17], improving the vascular integrity and decreasing the vascular permeability [21] and scavenging free radicals and protecting the stomach cell walls [22]. Bergamot flavonoids also exert anti-inflammatory activity with analgesic effects [23], protecting against high blood pressure and cholesterol in the blood [24,25,26,27]. 

Bergamot by-products, if transformed into flour, can be utilized as a functional ingredient in foods with a low or no significant antioxidant asset, such as baked goods. The *Pastazzo* flour can make these products healthier and may be considered a technological adjuvant, useful for improving and extending shelf-life. As a replacement for flour or bran in baked goods, it can increase the content of antioxidants, pectins and fibers. Fibers and pectins can promote a reduction in fats in recipes and counteract the staling process, thanks to the ability to retain water; they give greater stability to the product, even when subjected to freezing and thawing cycles [28]. Antioxidants are able to protect the product from stress, such as that generated by cooking. Moreover, it can have an antimicrobial activity and increase shelf life.

Additionally, the wheat flour is an important ingredient. Italian Law lists five different types of wheat flours produced with *Triticum aestivum* L. (Poaceae), classified according to protein content calculated as follows (N_2_ * 5.70): 00 type (minimum 9.0); 0 type (minimum 11.0); 1-, 2- and farina integrale di grano tenero (wholemeal soft wheat flour) types (minimum 12.0). Other physical values considered are the humidity and ash content which are, respectively, 14.5% on the total weight and a maximum of 0.55% on dry weight for 00 type wheat flour [29].

The functionalization of bakery products has been studied from many points of view [30,31,32,33,34,35,36,37,38]. Experiments have been conducted on the partial substitution of wheat flour in biscuits and in other bakery products to ameliorate the potential benefits in human health by incorporating: red lentil flour [39], common bean and maize flour [40], orange peel powder [41] and amaranth flour [42].

This is the first study evaluating how to use bergamot *Pastazzo* flour in a recipe for baked goods to ameliorate the chemical and physical properties of biscuits, allowing them to be considered as a functional food.

## 2. Materials and Methods

### 2.1. Raw Material

The bergamot fruits (BF) were processed by eliminating the flavedo and the juice during the crop season 2017/2018. The *Pastazzo* was pressed with a filter press before being dried at 45 °C by a dryer producing a tangential air flow (Scirocco model, Società Italiana Essiccatoi, Milan, Italy), until 12.6% of residual moisture content was reached. The dried bergamot *Pastazzo* was finely powdered (30 μm) to obtain a flour (BPF) able to be used as an ingredient (Figure 1).

### 2.2. Preparation of Biscuits

Four different formulations of biscuit were prepared and compared to a control sample. The basic biscuit ingredients (Control, C) were wheat flour 00 type, sugar, olive oil, eggs (whole) and baking powder (mixture of bicarbonate and weak acid). Biscuits were prepared without water in the dough. These ingredients were well kneaded with a laboratory mixer (Bimby TM31, Vorwerk, Wuppertal, Germany) and the dough was rolled out in 6 cm diameter and 3 mm thick discs. For biscuit preparation, a rolling pin equipped with a thickness measurement was used.

The control sample contained 100% wheat flour, while for the other four samples an aliquot of the 48 g wheat flour was replaced with four different percentages of BPF: 2.5% (BPF_2.5_), 5% (BPF_5_), 10% (BPF_10_) and 15% (BPF_15_). The five formulations varied only in the BPF included (Table 1). Biscuits with more than 15% BPF were not accepted by consumers. The biscuits were baked in an industrial oven at 180 °C for 12 min for the control biscuits and 8 min for the enriched biscuits. After cooking, each biscuit had a 7 cm diameter and a 1 cm thickness. Moreover, on the biscuits, the spread ratio was calculated by dividing the diameter by thickness of biscuits, showing values of about 7 (D/T) [43].

### 2.3. Characterization of Physical-Chemical Properties of Bergamot Pastazzo Flour

#### 2.3.1. Color Measurement in the CIE *L* a* b** System

The color analysis was performed by a tristimulus instrument (Minolta CR 300, Osaka, Japan). The CIE coordinates revealed by a D65 illuminant were: *L** (black-white), *a** (green-red) and *b** (blue-yellow). Chroma (C*) parameter represents the degree of saturation or fullness of color, and it was calculated with the following formula:C = (*a*^2^ + *b*^2^)^1/2^,

#### 2.3.2. Determination of pH, Moisture and Water Activity (a_w_)

The AACC International Method 02-52.01 was applied for pH analysis. In a beaker were placed: 15 g of sample and 10 mL of deionized water before being homogenized (30 min), then the mixture was left to decant (15 min). pH was measured with a pHmeter (Crison Basic 20, Barcelona, Spain).

The moisture content (MC) was determined as follows, 30 g of BPF sample was placed in an oven at 105 °C until a constant weight was reached. This determination was conducted in triplicate.

The a_w_ was determined on BPF by a hygrometer (Aqualab Model series 3TE, Decagon Devices, Inc. Pullman, Washington, DC, USA) previously calibrated. The analyses were conducted in three replicates.

#### 2.3.3. Determination of Total Phenolic Content

Total phenolic content (TPC) was determined according to the method reported by González Molina et al. [44], with some modifications. The analysis was performed on a 5 g sample. The analysis is more detailed in Section 2.5.2.

#### 2.3.4. Determination of Total Flavonoid Content

Total Flavonoid Content (TFC) was carried out following the method reported by Da Pozzo [16], with some modifications. The analysis was conducted on a 5 g sample. The analysis is more detailed in Section 2.5.3.

### 2.4. Characterization of Physical-Chemical Properties of Biscuits

#### 2.4.1. Determination of pH, Moisture and Water Activity (a_w_)

The pH determination was performed in accordance with the AACC International Method 02-52.01. In a beaker were placed: 15 g of ground biscuits and 10 mL of deionized water before to homogenized (30 min), then the mixture was left to decant (15 min). pH was measured with a pHmeter (Crison Basic 20, Barcelona, Spain).

The moisture content (MC) was determined by the gravimetric method, 5 g of ground biscuit sample was placed in an oven at 105 °C until a constant weight was reached. The results were expressed as mean of three replicates.

The a_w_ was determined on ground biscuits by a hygrometer (Aqualab Model series 3TE, Decagon Devices, Inc. Pullman, Washington, DC, USA) previously calibrated. The analyses were conducted in three replicates.

#### 2.4.2. Color Measurement in the CIE *L* a* b** System

The color analysis was conducted by a tristimulus instrument (Minolta CR 300, Osaka, Japan). The CIE coordinates revealed by a D65 illuminant were: *L** (black-white), *a** (green-red) and *b** (blue-yellow). Chroma (C*) parameter represents the degree of saturation or fullness of color, and it was calculated with the following formula:C = (*a*^2^ + *b*^2^)^1/2^,

While Whiteness index was calculated with equation: WI = 100 − [(100-*L*)^2^ + (*a*^2^ + *b*^2^)]^1/2^

For each formulation, 12 biscuits were randomly chosen, and color measurement was done on two points of the upper biscuit surface and on two points of the lower surface.

#### 2.4.3. Physical Properties and Sensory Overall Acceptability of Biscuit Samples

The structural properties (hardness, g) of the biscuits were assessed by a three-point bending test (TPB) using a TA-XT plus texture analyzer (Stable Micro Systems, Godalming, Surrey, UK). TPB is a destructive test based on the application of a vertical force to obtain texturometric parameters. Each sample was placed on the two holders of the adapter and the cutting probe was lowered until it met the sample. The probe acts as a third point of contact, exerting increasing pressure until the product structure breaks. The maximum peak force was used to calculate the hardness value. 

The sensory overall acceptability evaluation was carried out to compare the functionalized biscuits with a control sample. The samples were evaluated by the panelists according to different factors, including: appearance, aroma and flavor, with the aim of assaying the Sensory Overall Acceptability. The qualitative assessment was conducted by a panel of 8 peoples (male and female, between 30 and 60 years old), recruited among the staff of the Food Science and Technology Unit of Reggio Calabria University with prior experience in sensory analysis. The judges were trained before the sessions to identify the attributes to be evaluate. A nine-point hedonic scale was used to evaluate the taste, smell, appearance, texture and overall acceptability of biscuit samples. The results were obtained through the median calculation

### 2.5. Evaluation of Antioxidant Properties

#### 2.5.1. Extraction of Antioxidant Compounds on the Biscuits

Antioxidant compound extraction was performed following the methods reported by Imeneo et al. [45], appropriately modified. A total of 5 g of milled biscuits were weighed and added to 20 mL of methanol, 2.5 mL of distilled water and 0.250 mL of HCl (37%), the mixture was stirred at 30 °C for 120 min. The hydroalcoholic phase was separated from the solid phase by centrifugation at 6000 rpm for 10 min at 4 °C (in a refrigerated centrifuge apparatus, NF 1200R, Nüve, Ankara, Turkey) and recovered inside a vial after filtration through membrane filters with pores of 0.22 µm diameter. The obtained extract was used for the evaluation of antioxidant properties.

#### 2.5.2. Determination of Total Phenolic Content

Total phenolic content (TPC) was determined according to the method reported by González Molina et al. [44], with some modifications. An aliquot equal to 350 µL of methanol extract were mixed with 5 mL of water and 1 mL of Folin–Ciocalteu reagent, the solution was incubated for 8 min (room temperature); after which, 10 mL of Na_2_CO_3_ (7.5%) and distilled water were added to reach a volume of 25 mL. The solution was incubated in the dark for 2 h. Absorbance was measured at 765 nm using a double-beam ultraviolet-visible spectrophotometer (8453 UV–Vis, Agilent, Waldbronn, Germany). A calibration curve was prepared with gallic acid at different concentrations and the results were expressed as mg gallic acid equivalents g^−1^ biscuits (mg GAE g^−1^).

#### 2.5.3. Determination of Total Flavonoid Content

Total Flavonoid Content (TFC) was carried out following the method reported by Da Pozzo [16], with some modifications. In brief, 1 mL of methanol extract, 4 mL of water and 0.3 mL of NaNO_2_ (5%) were mixed and incubated at room temperature for 5 min. After, 0.3 mL of AlCl_3_ (10%) was added and it was incubated for a further 5 min. Then, 2 mL of NaOH (1 M) solution was added, and lastly, distilled water was used to make up the volume to 10 mL. At the same time, a solution used as a blank was prepared. The solution was incubated for 15 min at room temperature and the absorbance was measured at 510 nm using a spectrophotometer. A calibration curve was prepared with catechin at different concentrations and the results were expressed as mg catechin equivalents g^−1^ of biscuits (mg CE 100 g^−1^).

#### 2.5.4. Total Antioxidant Capacity Assays

The ABTS^.+^ (2.2′-azinobis-3-ethylbenzothiazoline-6-sulfonate) radical test was carried out as described by Re et al. [46]. The radical was generated by mixing 7 mM of ABTS^.+^ and K_2_S_2_O_8_ 140 mmol and followed by storage in the dark at room temperature for 16 h; before use, the solution was diluted (1:80) with ethanol. The reaction mixture was prepared by mixing 50 µL of methanolic extract and 2450 µL of the ethanol solution of ABTS^+^. The absorbance was measured after 6 min in the dark using a double-beam ultraviolet-visible spectrophotometer (Perkin-Elmer UV-Vis λ2, Waltham, MA, USA) at 734 nm. The results were expressed as µmol Trolox equivalent 100 g^−1^ of biscuits (µmol TE g^−1^), after plotting the absorbance against a calibration curve (Trolox from 3 to 18 μmol).

DPPH radical scavenging assay was carried out as Brand-Williams et al. [47] describe. For the reaction 100 µL of methanolic extract and 2400 µL of the DPPH methanolic solution were mixed. The absorbance was measured after 5 min at 515 nm in the dark using a double-beam ultraviolet-visible spectrophotometer. The results were expressed as µmol Trolox equivalent 100 g^−1^ of biscuits (µmol TE g^−1^).

### 2.6. Identification and Quantification of Individual Flavonoids (IF)

Individual Flavonoids were chromatographically analyzed following the method reported by Romeo et al. [48], appropriately modified. The chromatographic system consisted in a UHPLC PLATINblue (Knauer, Berlin, Germany) equipped with a binary pump system. Knauer blue orchid C18 column (100 mm length × 2 mm i.d. × 1.8 µm particle size) coupled with a PDA-1 (Photo Diode Array Detector) PLATINblue (Knauer, Berlin, Germany) and Clarity 6.2 software.

Before the analysis, the extracts were filtered (0.22 μm nylon) and 5 μL injected into the UHPLC. The mobile phase consisted of: solvent A—water acidified with acetic acid (pH 3.10); solvent B—acetonitrile. The stepwise gradient profile was: 0–3 min 5% B; 3–15 min 5–40% B; 15–15.5 min 40–100% B. At the end, initial conditions were restored during the run by keeping the column at 30 °C. For the quantification of each compound, external standards were used (concentration between 1 and 100 mg kg^−1^). The results were expressed as mg g^−1^.

### 2.7. Statistical Analysis

Samples were analyzed in triplicate with the exception of the determination of color (24 determinations) and hardness (10 determinations). Analytical data were reported as means ± standard deviation. The analysis of variance (one-way ANOVA) was conducted by applying the post hoc Tukey test at *p* < 0.05 (SPSS software, 25.0 version, Armonk, NY, USA). The following symbols were used to indicate the significance: *, *p* < 0.05; **, *p* < 0.01; ***, *p* < 0.001; *p* > 0.05, n.s.—not significant.

## 3. Results

The physical-chemical and antioxidant properties of BPF are reported in Table 2: the moisture content was 12.6% and the pH was 3.5. The high total polyphenol content was very relevant (19.5 mg gallic acid g^−1^). Other authors have studied the total phenolic content in the peel of different Citrus species and found: 23 mg GAE g^−1^ DW and 28.3 mg GAE g^−1^ DW in Indian *Citrus reticulata* cv. Kinnow mandarin [49]; 3900, 5900 and 5500 (mg GAE·kg^−1^ DW), respectively, for Spanish *Citrus sinensis* L., *Citrus lemon* L. and Citrus × Clementine [50].

### DPPH and ABTS: Total Antioxidant Activity Assays

Flavonoids are one of the most important fractions of bioactive molecules contained in citrus fruits. Studies have been conducted on flavonoids in citrus peel as an underutilized part of the fruit [51,52], and they were found to be agents for cancer prevention [53,54] and anti-virus treatments, in particular Hesperidin [55,56]. The individual flavonoids detected in BPF are reported in Table 2. The major compound was Hesperedin (113.33 mg g^−1^ DW) accounting for almost half of the total flavonoid content (TF: 4.57 mg 100 g^−1^ DW). The second and the third major compounds were Naringin (91.26 mg g^−1^) and Neoeriocitrin (53.35 mg g^−1^), by and large in the same ratio as their content in the bergamot juice [57]. Neohesperedin and Naringenin were 17.74 and 10.53 mg g^−1^, respectively. Neohesperedin in BPF showed a lower content compared to Neoeriocitrin (Table 2), a different behavior was described in the bergamot juice in which the latter was noticed to be in a higher content than in the former [57]. Narirutin and Eriocitrin in BPF accounted for less than 5 mg g^−1^ (Table 2).

After analytical characterization, the BPF was used for the formulation of functionalized biscuits. The study involved the five different types of biscuits described in Table 1 and produced with different percentages of bergamot *Pastazzo* flour replacing part of the wheat flour (BPF), ranging from 0% (control) to 15%. During preparation, the biscuit dough of samples BPF_5_, BPF_10_ and BPF_15_ was greasy suggesting the possibility to reduce the percentage of fat, in the presence of more than 5% of BPF.

Color is an important parameter which influences consumer acceptance of food in general [58,59,60] and of biscuits in particular [61,62]. The biscuits enriched with BPF reached the desired color in a shorter time, probably because the BPF was more susceptible to browning compared to the flour of wheat. The Maillard non-enzymatic browning reaction is influenced by many factors such as: temperature, sugars, a_w_, pH, type and amino compounds [63,64].

The findings related to the color analysis of biscuits prepared with BPF are shown in Table 3.

A peculiar characteristic of baked products is the color which varies during baking. If the upper surface values are compared with the lower surface ones, the color parameters showed an inverse behavior in the studied cookies. In fact, no significant differences were found by comparing the *b** and C* values of the upper surface samples, whereas very high significant differences (*p* < 0.001) were found by observing the same parameters in the lower surface of biscuits with a tendency to increase: the higher the BPF content in the dough, the higher the *b** and C* values. In contrast, *L** differed significantly in the upper surface of the five samples, whereas it showed no significant differences in the lower surface. *a** showed significant differences (*p* < 0.05) in the lower surface and very high differences (*p* < 0.001) in the upper surface (Table 3). The behavior of *a** was also explained in a study conducted by Jan et al. [65], in which it was found that the partial replacement of wheat flour with buckwheat flour reduced the redness in cookies.

An increase in brightness was detected on the upper surface, corresponding to a lower black component content. The lower browning detected in the products obtained with the addition of BPF highlighted lower values of the parameter *a**. This may be due to a reduction in the ‘red’ component typical of non-enzymatic browning. 

Statistically significant differences were observed mainly in the upper surface of the biscuits. In fact, with the addition of BPF in lower concentrations, there is an increase in the *L** parameter and a reduction in the red component (*a**). With the increase in the concentration of BPF, again on the upper side, there is an increase in the red component compared to the BPF2.5 sample. The chromatic reliefs developed on the lower face are affected by a transmission of heat by conduction with a surface temperature higher than that of the air used for cooking and in contact with the upper surface.

Al Saab studied biscuits prepared functionalized cookies and according to 100 g wheat flour basis, 0, 5, 15 and 20% orange peel powder were used to replace wheat flour. They observed a reduction in *L** from 72.68 (control, 100% wheat flour), to 65.42, 62.07, 58.48 and 57.11, respectively, for the four replacements. At the same time, *a** increased from 0.54 (control) to 5.69 (20% orange peel powder replacement) and *b** increased from 25.62 (control) to 32.66 (20% orange peel powder replacement) [41]. 

Gurram and Sharma studied biscuits prepared with wheat and pearl millet flour and fortified with orange peel powder (from 3 to 15% replacement) and found that the higher the orange peel powder content, the lower the *L** value, from 91.57 (3% orange peel powder) to 53.85 (15% orange peel powder) [66].

Kozlowska et al. [67] studied shortbread cookies prepared with different herbal extracts (Hyssop, Lemon balm and Nettle) and found that both the type of extract and its concentration in the recipe highly influenced *L**, *a** and *b** parameters of biscuits.

The higher the BPF content, the lower the pH value (*p* < 0.001); this was due to the effect of the pH of the BPF on the recipe (Table 4).

The a_w_ values are reported in Table 4. The a_w_ in biscuits is influenced by the environmental temperature and increases with the temperature increase from 20 to 30 °C [68].

The a_w_ value increased significantly (*p* < 0.001) with the increase in the concentrations of BPF in biscuits between 0.228 (C) and 0.281 (BPF15) as shown in Table 4, this is related with the different fiber and pectins content in BPF in fact their hygroscopicity causes the absorption and retention of water in foods [69]. Coherently with the increase in a_w_ values of biscuits, the moisture content values also showed a slight increase, even if statistically not significant, with the increase in the quantity of BPF. Our results are in agreement with findings of other authors who studied bakery products prepared with apple fiber and found that the increase in apple fiber in the recipe increased the water content in bakery products [70]. The relationship between fiber addition in the recipe and moisture retaining in biscuits after cooking was also confirmed in a study conducted on biscuits functionalized with lemon pomace, also in this work the authors have found the highest moisture content in the biscuits prepared with the highest lemon pomace content [45]. 

Table 5 shows the results relating to physical properties, particularly the hardness values. The biscuits’ texture was significantly different among the samples. The control sample showed the lowest hardness value (1823.15 g), while, with an increase in the concentration of BPF, the hardness also increased. In fact, the biscuits produced with 5, 10 and 15% of bergamot *Pastazzo* flour (BPF) showed higher values of hardness (2637, 2717 and 2817, respectively). The increase may be due to an interaction in the formation of the gluten mesh which, as a result of the fiber present in the pulp, can stiffen with the presence of components other than flour. An inverse behavior was found in biscuits prepared with wheat flout partially substituted with roasted flaxseed flour (0, 10, 25 and 40%); here, the hardness decreased with the increase in the roasted flaxseed flour; this could be due to the different composition of the flour studied in this work, in particular in relation to the high oil content in the roasted flaxseed flour [71].

The consumer acceptability is one of the most important requirements when a food product has to be offered to the market. For this reason, the cooked biscuits were evaluated by a group of panelists. Table 5 reports the values related to the overall sensory acceptability of the biscuits. Among the different concentrations of BPF added to the biscuits, the panelists preferred those obtained with the lower concentration (BPF2.5, about 7.15) also when compared to the control sample, which showed moderate acceptability (6.45). A lower acceptance of the other samples was observed, particularly those formulated with a higher concentration of BPF (10 and 15%), which revealed a strong bitterness. In accordance with our results, other authors have demonstrated the effect of flour composition on perception and overall acceptability of consumers and how the excessive content of a functional ingredient could decrease consumer acceptability [65,72].

The functionalization of biscuit samples with BPF allowed us to obtain newly formulated products with an enhanced antioxidant capacity. Indeed, the evaluation of antioxidant capacity is a fundamental parameter to understand the real importance of this type of application in the food system. 

Antioxidant assays showed important results, as reported in Table 6. TPC and TF values in the functionalized samples tended to increase as the BPF content in the biscuit formulation also increased (*p* < 0.001).

The TPC value tended to significantly increase in the functionalized biscuits, in particular, already the addition of 2.5% BPF leads to an increase in TPC to 0.60 mg GAE g^−1^, more than four times compared to the control (0.14 mg GAE g^−1^). All the functionalized samples showed a high TPC value, which grows almost proportionally to the BPF added. Indeed, the maximum TPC value was tested in BPF15 (3.64 mg GAE g^−1^). Al-Saab and Gdallah [18] studied cookies fortified with orange peel powder (OPP) and found the following increase in total phenolic content: 1.15 mg GAE g^−1^ (control without OPP), 5.54 mg GAE g^−1^ (5% OPP), 6.10 mg GAE g^−1^ (10% OPP), 8.31 mg GAE g^−1^ (15% OPP) and 9.12 mg GAE g^−1^ (20% OPP). This confirms our results: the higher the citrus peel content, the higher the phenolic content. 

Imeneo et al. prepared biscuits with lemon by-products and found: a TPC of 0.59 mg GAE g^−1^ d.w. in the lemon pomace extract; 12 and 15 mg GAE g^−1^ d.w, respectively, in the dough and in the biscuits used as control (without lemon by-products); 28 and 30 mg GAE g^−1^ d.w in the dough and biscuits prepared with fresh lemon peel; 29 and 32 mg GAE g^−1^ d.w in the dough and biscuits prepared with fresh lemon peel and lemon pomace extracts; 28 and 34 mg GAE g^−1^ d.w, respectively, for dough and biscuits prepared with lemon pomace extracts. In this work was also evidenced as the antioxidant activity measured with the DPPH assay was influenced by the different lemon by-products and as the dough had DPPH values higher than the corresponding biscuits [45]. 

Al-Saab and Gadallah studied cookies functionalized with orange peel powder (5, 10, 15 and 20%) to substitute wheat flour. They found a TPC of 1.15 mg GAE/g in the control biscuits (100% wheat flour) and 5.54, 6.10, 8.31 and 9.12 mg GAE/g, respectively, in the functionalized cookies. The scavenging activity measured with the DPPH assay was: 2.65% (control) and 4.55, 17.48, 25.25 and 40.92%, respectively for the functionalized cookies [41].

Olcay and Demir studied biscuits prepared with Kumquat fruit flour in replacement of 10, 20 and 30% of wheat flour and found a TPC increase from 746.18 μg GAE g^−1^ (control, without Kumquat fruit flour) to 1223.90 μg GAE g^−1^ (10% replacement); 1618.63 μg GAE g^−1^ (20% replacement) and 2080.10 μg GAE g^−1^ (30% replacement). At the same time the highest overall acceptability was found in the 10% wheat flour replacement. This confirmed our results that the citrus flour addition in the biscuit recipe increases the phenolic content but a citrus powder content that is too high decreases the overall acceptability of consumers [73].

Improved results were found also for biscuits functionalized with flour of non-citrus origin. Akubor and Owuse studied biscuits prepared with wheat flour (80%) and functionalized with tomato peel flour (20%) and found: 1082.22 mg/100 g TPC in tomato peel flour, 4.4 mg/100 g TPC in wheat flour and 123.39 mg/100 g TPC in the so functionalized biscuits; it means that in that case a potential TPC increase of about 215 mg/100 g produced, de facto, an increase of about 122 mg/100 g TPC. This loss in TPC could be due to the baking effect [74]

Additionally, the TFC analysis showed the same trend, with increasing values in the functionalized biscuits (BPF15 > BPF10 > BPF5 > BB2.5).

The TFC for BPF10 and BPF15 calculated with the spectrophotometric method (Table 6) showed a different result if compared with the sum of single flavonoids obtained with the HPLC (Table 7), this was due to the two different methods applied.

As per TPC and TFC, also the antioxidant activity of the BPF functionalized biscuits, measured by ABTS and DPPH assays, was found to be significantly higher than in the control (Table 6). ABTS assay highlighted significant variation among the samples with higher values for BPF10 and BPF15 (63.10 and 64.24 mg TE 100 g^−1^, respectively). It has to be pointed out that the increase in percentage from BPF10 to BPF15 showed a non-linear increase in the ABTS value. For the DPPH assay, the relationship is not very clear; indeed, there was an increase in DPPH value in all the functionalized biscuits compared to the control, but a linear increase in the DPPH value. It must be emphasized that the two tests have different molecules as reagent/target and consequently the responses to the assays are also different. In addition, the different antioxidants contained in citrus juice were found to have a synergistic effect, for this reason, it is strongly suggested the use of more than one assay [57,75]. The ABTS assay value of the BPF15 increased but not in a linear progression with TPC and TFC increase, this was in accordance with results of other authors in studies related to citrus juices and it is probably due to the synergistic effect existing when different phenolic compounds are considered in a matrix, in fact, even if the TPC or the TFC increase with the increase in a variable, the single phenols do not always increase linearly and do not contribute equally to the sum of TPC and TFC [76]. This could be due to the different behavior of different phenolic molecules and the baking temperature.

In partial disagreement with our results, Al-Saab and Gadallah [29] found a positive correlation between the DPPH assay and the orange peel powder fortification of cookies.

Ojha and Tapa studied biscuits incorporated with mandarin peel powder. They reported that a 6% mandarin peel in the recipe increased the TPC from 1.32 mg GAE g^−1^ (control) to 128.92 mg GAE g^−1^ (functionalized biscuits), and at the same time as the antioxidant activity measured with the DPPH assay increased from 23.72% in the control to 73% in the functionalized biscuits [77].

Obafaye and Omoda incorporated 5, 10, 15 and 20% orange peel flour in pearl-milled flour to prepare functionalized biscuits and evidenced an increase in TPC from 5.84 mg GAE g^−1^ (control, 100% pearl milled flour) to 8.32, 9.60, 10.32 and 118.87 mg GAE g^−1^, respectively, for the different percentages of functionalized biscuits. They found that the biscuits prepared with the highest orange peel flour content (20%) showed the highest ABTS value (2.19 mMol TEAC g^−1^) related with the highest TPC (1.87 mg GAE g^−1^) and the highest TFC (8.12 mg QE g^−1^). The biscuits prepared without orange peel flour contained the lowest TPC (5.84 mg GAE g^−1^), the lowest TFC (1.20 mg QE g^−1^) and the lowest ABTS value (1.17 mmol TEAC g^−1^) [78]. 

Findings of Fernández-Fernández et al. related to the in vitro bioactivity of the bioaccessible fractions of biscuits, revealed as both digested and undigested samples of biscuits prepared with Navel and Valencia varieties oranges pomace (10% calculated on the total ingredients) showed antioxidant capacity values higher than the control biscuits prepared without citrus pomaces. At the same time, both ABTS and ORAC-FL values calculated in the functionalized biscuits during the digestive process, appear to have a bioactive protection against degradation during the digestive process [79].

The principal identified and quantified flavonoids in the enriched biscuits are the same detected for the BPF and the results are reported in Table 7.

In the biscuits functionalized with BPF, seven individual flavonoid compounds (IF) were identified and quantified. The individual flavonoid content was found to be linear in the range BPF2.5 and BPF10, whereas it was found to be non-linear with BPF15. Hesperidin showed the highest content ranging from 0.25 to 4.93 mg g^−1^, followed by Naringin (0.21 to 3.88 mg g^−1^). The least represented flavonoid was Eriocitrin ranging from 0.003 to 0.10 mg g^−1^. The 2.5 to 5% BPF variation in the recipe produced a double total flavonoid content in the biscuits, from 0.64 to 1.27 mg g^−1^. The highest flavonoid content (12.89 mg g^−1^) was found in the biscuits fortified with 15% BPF, it means that a 30 g portion of BPF fortified biscuits (3 biscuits) contains 387 mg of flavonoids.

It has to be stated that the BPF addition in the recipe in the range between 2.5% and 10% resulted in a liner increase for TPC and TFC spectrophotometrically measured and also for the individual flavonoid detected by HPLC, whereas the BPF_15_ addition resulted in a more than proportional increase. This could be due to the excessive BPF content in the recipe.

The high flavonoid content of the newly formulated biscuits confirms the importance of the use of natural antioxidants (contained in BPFs). The consumption of functionalized products could have an important health value for the consumer’s diet.

Our data are corroborated by findings of other authors on biscuits functionalized with citrus fruits and their derivatives. Imeneo et al. used fresh lemon peel and lemon pomace extract to prepare functionalized biscuits and found that the addition of functional compounds (mainly Eriocitrin and Hesperidin) to biscuits increased the antioxidant properties in doughs and at the same time protected the samples during the thermal treatment of cooking [45].

Table 8 depicts the correlation matrix between some physical-chemical-sensory parameters of the BPF fortified biscuits. pH showed a strong negative correlation with all parameters varying between −0.681 (DPPH) and −0.976 (TPC).

TPC showed a strong positive correlation with TFC (*r* = 0.994; *p* = 0.544; R^2^ = 0.988; *t* = 0.634) and with ABTS (*r* = 0.933; *p* = 0.003; R^2^ = 0.870; *t* = 4.266), evidencing that these three parameters are highly correlated. ABTS and DPPH assays were characterized by a positive correlation (*r* = 0.830; *p* = 0.000; R^2^ = 0.689; *t* = 39.212), but it has to be pointed out that DPPH values showed a non-linear increasing trend, in particular the value of BPF2.5 was higher than the ones of other recipes, this confirm that, in this case, the ABTS assay has to be preferred to DPPH assay. 

TFC showed a low correlation with DPPH values (*r* = 0.566; *p* = 0.209; R^2^ = 0.321; *t* = 1.365) due to an ineffective response of this assay in this context.

Additionally, TPC showed a low correlation with DPPH values (*r* = 0.651; *p* = 0.299; R^2^ = 0.423; *t* = 1.110) confirming an ineffective response of this assay in this context.

The aw parameter was strongly and positively correlated with the moisture content (*r* = 0.906; *p* = 0.000; R^2^ = 0.821; *t* = 26.352). 

SOA was negatively correlated with TFC (*r* = −0.958; *p* = 0.01; R^2^ = 0.918; *t* = 3.240) and with all the biscuit antioxidant activity parameters. This was due to the high impact of bergamot flour on the sensory perception of panelists. This is also well explained in Table 5, which reports that the panelists preferred the BPF_2.5_ and that, although an increase in the recipe’s BPF content increased the biscuits’ functionalization, it decreased their acceptability to the panelists. 

## 4. Conclusions

The performed analyses evaluated the physical-chemical-sensory quality of biscuits fortified with bergamot *Pastazzo* flour, added in different percentages to partially replace the wheat flour. The addition of Bergamot *Pastazzo* flour caused an increase in water activity without reaching values that could make the product susceptible to microbiological alteration. The pH undergoes a decrease for all samples with BPF; compared to the control, this was more evident for the BPF samples at 10 and 15%. The antioxidant capacity can be evaluated by both ABTS and DPPH assay. The higher the quantity of bergamot *Pastazzo* flour in the biscuit recipe, the higher the polyphenol and flavonoid content with hesperidin and naringin as the major components. Bergamot *Pastazzo* flour can be considered an ingredient to obtain a functional food. The enriched samples required a shorter cooking time which reduced the time of exposure to thermal stress and the production cost. The 2.5% bergamot *Pastazzo* flour content in the biscuit recipe was the quantity preferred by the panelists in terms of overall acceptability. 

## Figures and Tables

**Figure 1 foods-11-01137-f001:**
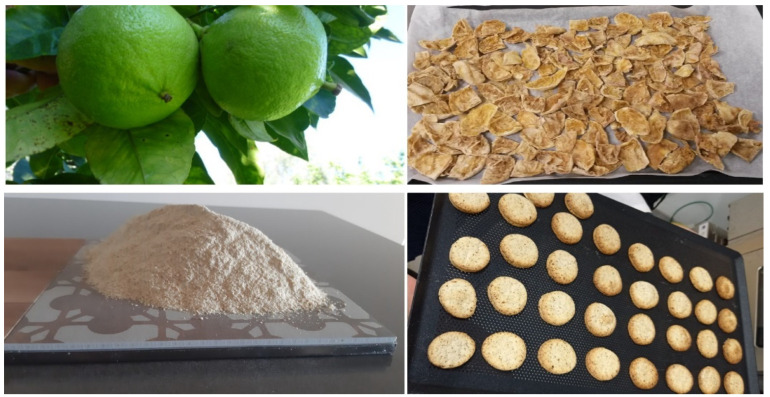
From bergamot fruits to functionalized biscuit production.

**Table 1 foods-11-01137-t001:** Sample composition (g 100 g^−1^ of sample) and denomination of biscuits.

Samples	Wheat Flour (g)	Bergamot *Pastazzo* Flour (g)	Sugar (g)	Extra VirginOlive Oil (g)	Eggs (g)	Baking Powder (g)
C	48	0	24	17	10	1
BPF_2.5_	46.8	1.2	24	17	10	1
BPF_5_	45.6	2.4	24	17	10	1
BPF_10_	43.2	4.8	24	17	10	1
BPF_15_	40.8	7.2	24	17	10	1

BPF: 2.5% (BPF_2.5_), BPF: 5% (BPF_5_), BPF: 10% (BPF_10_) and BPF: 15% (BPF_15_).

**Table 2 foods-11-01137-t002:** Physical-chemical, antioxidant properties and concentration values of Individual Flavonoids (IF) in Bergamot *Pastazzo* Flour (BPF).

*L**	60.18 ± 1.11
*a**	7.62 ± 0.64
*b**	22.15 ± 2.44
a_w_	0.416 ± 0.051
Moisture (%)	12.6 ± 1.2
pH	3.5 ± 0.2
TPC (mg GAE g^−1^)	19.53 ± 0.13
TF (mg CE 100 g^−1^)	4.57 ± 0.04
Eriocitrin (mg g^−1^)	1.58 ± 0.26
Neoeriocitrin (mg g^−1^)	53.35 ± 3.65
Narirutin (mg g^−1^)	4.13 ± 0.62
Naringin (mg g^−1^)	91.26 ± 8.46
Hesperedin (mg g^−1^)	113.33 ± 10.71
Neohesperedin (mg g^−1^)	17.74 ± 1.95
Naringenin (mg g^−1^)	10.53 ± 0.59

*L**: lightness (black-white); *a**: green-red; *b**: blue-yellow; a_w_: water activity; GAE: Gallic acid equivalent; TPC: Total Phenolic Compounds; CE: Catechin Equivalents; TF: Total Flavonoids.

**Table 3 foods-11-01137-t003:** Upper and lower surface color of the biscuit.

	Samples	*L**	*a**	*b**	C*	WI
Upper Surface	C	62.14 ^a^	8.94 ^c^	26.55	28.08	52.90
	BPF_2.5_	71.88 ^c^	5.25 ^ab^	29.07	29.58	59.21
	BPF_5_	72.09 ^c^	4.56 ^a^	25.90	26.32	61.64
	BPF_10_	67.49 ^b^	6.44 ^b^	28.67	29.41	56.04
	BPF_15_	68.66 ^b^	6.29 ^ab^	27.90	28.63	57.57
	Sign.	***	***	n.s.	n.s.	n.s.
Lower Surface	C	68.50	6.15 ^a^	25.75 ^a^	26.55 ^a^	58.85
	BPF_2.5_	67.60	7.89 ^b^	27.84 ^b^	28.99 ^b^	56.55
	BPF_5_	69.22	6.97 ^ab^	28.34 ^b^	29.24 ^b^	57.58
	BPF_10_	67.44	6.05 ^a^	28.65 ^b^	29.35 ^b^	56.20
	BPF_15_	66.28	7.64 ^b^	28.78 ^b^	29.58 ^b^	55.01
	Sign.	n.s.	*	***	***	n.s.

The data are presented as means ± SD (*n* = 3). Means within a column with different letters are significantly different by Tukey’s post hoc test. Abbreviations: * significance at *p* < 0.05; *** significance at *p* < 0.001; n.s.—not significant. BPF—Bergamot *Pastazzo* Flour.

**Table 4 foods-11-01137-t004:** Water activity (a_w_), Moisture (%) and pH of the biscuit samples.

Samples	pH	a_w_	Moisture (%)
C	7.14 ^c^	0.228 ^b^	4.35
BPF_2.5_	6.79 ^b^	0.213 ^a^	4.32
BPF_5_	6.48 ^b^	0.270 ^d^	4.78
BPF_10_	6.00 ^a^	0.257 ^c^	4.54
BPF_15_	5.94 ^a^	0.281 ^d^	5.20
Sign.	***	***	n.s.

The data are presented as means ± SD (*n* = 3). Means within a column with different letters are significantly different by Tukey’s post hoc test. Abbreviation: *** significance at *p* < 0.001; n.s.—not significant.

**Table 5 foods-11-01137-t005:** Physical properties and sensory overall acceptability of biscuit samples.

Samples	Hardness Value (g)	Sensory Overall Acceptability (SOA)
C	1823.15 ^a^	6.45 ^b^
BPF_2.5_	2021.95 ^b^	7.15 ^a^
BPF_5_	2637.45 ^c^	5.85 ^c^
BPF_10_	2717.22 ^c^	4.05 ^d^
BPF_15_	2817.57 ^c^	3.45 ^e^
Sign.	***	***

The data are presented as means ± SD (*n* = 3). Means within a column with different letters are significantly different by Tukey’s post hoc test. Abbreviation: *** significance at *p* < 0.001.

**Table 6 foods-11-01137-t006:** Total antioxidant capacity (TPC, TFC, ABTS and DPPH) of the biscuits samples.

Samples	TPC (mg GAE g^−1^)	TFC (mg CE g^−1^)	ABTS (mg TE 100 g^−1^)	DPPH (mg TE·100 g^−1^)
C	0.14 ^a^	n.d.	10.67 ^a^	129.40 ^a^
BPF_2.5_	0.60 ^b^	0.64 ^a^	37.84 ^b^	1318.48 ^d^
BPF_5_	1.14 ^c^	1.27 ^b^	49.14 ^c^	1181.81 ^b^
BPF_10_	2.00 ^d^	2.68 ^c^	63.10 ^d^	1178.14 ^b^
BPF_15_	3.64 ^e^	3.90 ^d^	64.24 ^e^	1244.41 ^c^
Sign.	***	***	***	***

The data are presented as means ± SD (*n* = 3). Means within a column with different letters are significantly different by Tukey’s post hoc test. Abbreviation: n.d. = not detected. *** significance at *p* < 0.001.

**Table 7 foods-11-01137-t007:** Concentration values of Individual Flavonoids (IF, mg g^−1^) in Biscuits.

Compounds	C	BPF2.5	BPF5	BPF10	BPF15	Sign.
Eriocitrin	n.d.	n.d.	0.003 ^a^	0.07 ^b^	0.10 ^c^	***
Neoeriocitrin	n.d.	0.12 ^a^	0.23 ^b^	1.53 ^c^	2.33 ^d^	***
Narirutin	n.d.	0.005 ^a^	0.006 ^b^	0.06 ^c^	0.09 ^d^	***
Naringin	n.d.	0.21^a^	0.41 ^b^	2.69 ^c^	3.88 ^d^	***
Hesperedin	n.d.	0.25 ^a^	0.49 ^b^	3.41 ^c^	4.93 ^d^	***
Neohesperedin	n.d.	0.02 ^a^	0.06 ^b^	0.30 ^c^	0.53 ^d^	***
Naringenin	n.d.	0.03 ^a^	0.07 ^b^	0.56 ^c^	1.03 ^d^	***

The data are presented as means ± SD (*n* = 3). Means within a column with different letters are significantly different by Tukey’s post hoc test. Abbreviation: n.d. = not detected; *** significance at *p* < 0.001.

**Table 8 foods-11-01137-t008:** Correlation matrix.

	pH	a_w_	Moisture	Hardness	Total Polyphenols	Total Flavonoids	ABTS	DPPH	SOA
pH	1	*−0.786* 0.000	*−0.730* 0.000	*−0.950* 0.000	*−0.976* 0.002	*−0.958* 0.011	*−0.971* 0.000	*−0.681* 0.000	*−0.945* 0.012
a_w_	0.618 33.298	1	*0.906* 0.000	*0.908* 0.000	*0.772* 0.002	*0.792* 0.012	0.725 0.000	*0.350* 0.000	*−0.824* 0.000
Moisture	0.533 6.437	0.821 26.352	1	*0.801* 0.000	*0.802* 0.002	*0.828* 0.012	*0.679* 0.000	*0.409* 0.000	*−0.764* 0.607
Hardness	0.902 15.847	0.825 15.885	0.642 15.857	1	*0.902* 0.057	*0.884* 0.545	*0.941* 0.000	*0.655* 0.000	*−0.872* 0.000
Total polyphenols	0.953 4.394	0.596 4.410	0.643 4.398	0.813 2.220	1	*0.994* 0.544	*0.933* 0.003	*0.651* 0.299	*−0.916* 0.002
Total flavonoids	0.918 3.240	0.628 3.250	0.686 3.243	0.780 0.632	0.988 0.634	1	*0.891* 0.013	*0.566* 0.209	*−0.945* 0.012
ABTS	0.943 8.426	0.526 9.509	0.461 8.711	0.885 15.543	0.870 4.266	0.794 3.168	1	*0.830* 0.000	*−0.780* 0.000
DPPH	0.463 41.517	0.123 41.723	0.167 41.571	0.429 8.079	0.423 1.110	0.321 1.365	0.689 39.212	1	*−0.772* 0.000
SOA	0.893 3.242	0.679 6.435	0.584 0.535	0.716 15.854	0.839 4.397	0.893 3.242	0.608 8.563	0.100 41.544	1

In the South-West section of the matrix there are the R^2^ value (above) and the *t*-value (below and underlined) with the significance of the *t*-test calculated at 95% confidence interval. In the North-East section there are the *r* value (above and in italic font) and the significance level (below) calculated at 95% confidence interval. SOA is the abbreviation for Sensory Overall Acceptability. For TFC we used values of Table 6.

## Data Availability

Not available.

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
