# Peer review of "Formulation of Biscuits Fortified with a Flour Obtained from Bergamot By-Products (Citrus bergamia, Risso)"

_foods, 2022, doi:10.3390/foods11081137_

Round 1

Reviewer 1 Report

The comments are in the manuscript and highlighted in yellow. The manuscript requires a thorough revision. need to determine spread, WI etc

Author Response

Response to Reviewer 1

Thank you very much for your suggestions and changes requested, which have undoubtedly improved our paper.

All your comments have been included in the manuscript.

Regards.

Reviewer 2 Report

The paper wanted to evaluate the effect of addition bergamot pastazzo on quality of shortbread biscuits. There are some problems about the manuscript.  The questions from the manuscript is as following.

  • Background was poorly presented, what’s the total production of bergamot, how many tons of by product pastazzo can be produced? What’s the role of polyphenols and flavonoids contained in bergamot to human health?
  • The tables were poorly presented, several tables should be merged. The abbreviation in table should be note under the table. Maybe some table can be changed into figure.
  • please perform statistical analysis and put the corresponding letters in tables.
  • Diameter and thickness, also spread ratio are important traits of biscuit, the author should add these data.
  • In Table 10, the significant relation should be note.
  • There is no discussion.

Author Response

The paper wanted to evaluate the effect of addition bergamot pastazzo on quality of shortbread biscuits. There are some problems about the manuscript.  The questions from the manuscript is as following.

RESPONSE: Thank you very much for your suggestions and changes requested, which have undoubtedly improved our paper.

Background was poorly presented, what’s the total production of bergamot, how many tons of by product pastazzo can be produced? What’s the role of polyphenols and flavonoids contained in bergamot to human health?

  1. The effects of phenols and flavonoids on human health were discussed in the introduction section;

The tables were poorly presented, several tables should be merged. The abbreviation in table should be note under the table. Maybe some table can be changed into figure.

  1. tables are now merged, the abbreviations are explained. We have uses tables to better read values and also for the short time available.

please perform statistical analysis and put the corresponding letters in tables.

  1. The statistical analysis was performed and the corresponding letters are in tables

Diameter and thickness, also spread ratio are important traits of biscuit, the author should add these data.

  1. The required data are now included

In Table 10, the significant relation should be note.

  1. statystical data are now included

There is no discussion.

  1. the discussion was improved and included with the discussion of results.

Regards

Reviewer 3 Report

The manuscript presents the formulation of high antioxidant biscuits by partially replacing wheat flour with bergamot by-products. Data of phytochemical content and antioxidant activity is intensively reported. However, some methodology is not clear.

  1. Line 49-51, please report the content of those compositions.
  2. Line 84, Is it ‘humidity’ or ‘moisture content’?
  3. Please specify the particle size of BPF. Actually flour properties (such as viscoelasticity, starch content, etc) of BPF should be studied.
  4. Please give some reasons to design the amount of BPF in biscuit in the range 0-7.2 g. Any references?
  5. Did authors add water in the formular? If so, how much it it?
  6. Methodology in 2.3, 2.4 and 2.5 explained how to evaluate the biscuits, not flour. However, the results of flour characteristics were presented in Table 2 and 3. Please revise the methodology.
  7. Acceptability test should be tested with a large number of consumers or general panelists, rather than the trained panelists.
  8. How to separate upper surface and lower surface of biscuits for color measurement?
  9. Table 4 and 5 can be combined as color of biscuits.
  10. Table 2 and 3 can be combined as BPF flour characteristics.
  11. Line 262 ……there is an increase in the red component compared to the BPF2.5 sample…, However, Table 4 shows non-significant difference among the BPF biscuits. Please recheck the results.
  12. Line 258, please explain a bit more on how BPF reduce redness from the non-enzymatic browning reaction.
  13. Table 8, TPC was increased proportionally to the increased BPF in biscuits from 0-10%. However, TPC was dramatically increased, when 15%BPF was added. Please explain the reason of this observation.
  14. In contrast to No.13, antioxidant activity by ABTS assay was increased slightly when % BPF increased from 10 to 15%. Please clearly discuss on this finding.
  15. Table 9, why were the individual flavonoids in biscuits increased dramatically (not proportionally) when 10-15% BPF was added in biscuits?
  16. Line 346-347, please clearly discuss why the results of ABTS and DPPH were different.
  17. Line 345……but not there were no significant difference among the different used concentrations.

What does the sentence mean? Table 8 shows the significant difference among BPF biscuits.

Author Response

Response to Reviewer 3

Comments and Suggestions for Authors

The manuscript presents the formulation of high antioxidant biscuits by partially replacing wheat flour with bergamot by-products. Data of phytochemical content and antioxidant activity is intensively reported. However, some methodology is not clear.

RESPONSE: Thank you very much for your suggestions and changes requested, which have undoubtedly improved our paper.

  1. Line 49-51, please report the content of those compositions.
  2. done
  3. Line 84, Is it ‘humidity’ or ‘moisture content’?
  4. moisture content
  5. Please specify the particle size of BPF. Actually flour properties (such as viscoelasticity, starch content, etc) of BPF should be studied.
  6. done
  7. Please give some reasons to design the amount of BPF in biscuit in the range 0-7.2 g. Any references?
  8. The range 0-7.2 is due to the percentage of bergamot Pastazzo flour and it is calculated as percentage on 48 g wheat flour. We have conducted many tests before to verify that 15% bergamot Pastazzo flour is the maximum accepted by consumers.
  9. Did authors add water in the formular? If so, how much it it?
  10. No water was added. Now this is evidenced in the text.
  11. Methodology in 2.3, 2.4 and 2.5 explained how to evaluate the biscuits, not flour. However, the results of flour characteristics were presented in Table 2 and 3. Please revise the methodology.
  12. Methodologies are now completed.
  13. Acceptability test should be tested with a large number of consumers or general panelists, rather than the trained panelists.
  14. Thank you for your suggestion. We will apply this method in our next work.
  15. How to separate upper surface and lower surface of biscuits for color measurement?
  16. Surfaces were not separated: first the upper surface was analysed, secondly the lower surface of the biscuits was analysed;
  17. Table 4 and 5 can be combined as color of biscuits.
  18. done.
  19. Table 2 and 3 can be combined as BPF flour characteristics.
  20. done
  21. Line 262 ……there is an increase in the red component compared to the BPF2.5 sample…, However, Table 4 shows non-significant difference among the BPF biscuits. Please recheck the results.
  22. results re-checked
  23. Line 258, please explain a bit more on how BPF reduce redness from the non-enzymatic browning reaction.
  24. discussion improved;
  25. Table 8, TPC was increased proportionally to the increased BPF in biscuits from 0-10%. However, TPC was dramatically increased, when 15%BPF was added. Please explain the reason of this observation.
  26. discussion improved;

In contrast to No.13, antioxidant activity by ABTS assay was increased slightly when % BPF increased from 10 to 15%. Please clearly discuss on this finding.

  1. discussion improved;

Table 9, why were the individual flavonoids in biscuits increased dramatically (not proportionally) when 10-15% BPF was added in biscuits?

  1. discussion improved;

Line 346-347, please clearly discuss why the results of ABTS and DPPH were different.

  1. discussion included;

Line 345……but not there were no significant difference among the different used concentrations.

What does the sentence mean? Table 8 shows the significant difference among BPF biscuits.

  1. Corrections done.

Regards

Reviewer 4 Report

The argument is interesting and well treated, the experiment is well designed and data are well discussed. Some improvement is necessary. 1) Abstract section: please insert some relevant numeric value more; 2) Caption of figure 1. After the figure number, replace colons with a period; 3) References section, when you have written the titles of papers, sometime you have written in capital letter (Ref 2) and sometime in small letter (ref 3). Be consistent in the whole section and apply the instructions for authors of Foods; 4) 2.5 sub-section, when you list the column characteristics, replace width with internal diameter; 5) 2.6 sub section, line 220. Apply the same spacing between letter, symbol and numeric value when you indicate the significance. Sometime you have written p

Author Response

Response to Reviewer 4

RESPONSE: Thank you very much for your suggestions and changes requested, which have undoubtedly improved our paper.

Comment:

1) The argument is interesting and well treated, the experiment is well designed and data are well discussed. Some improvement is necessary.

  1. Thank you.

2) Abstract section: please insert some relevant numeric value more;

  1. Some relevant numeric value more was included.

 3) Caption of figure 1. After the figure number, replace colons with a period;

  1. Correction done.

 4) References section, when you have written the titles of papers, sometime you have written in capital letter (Ref 2) and sometime in small letter (ref 3). Be consistent in the whole section and apply the instructions for authors of Foods;

  1. Correction done.

5) 2.5 sub-section, when you list the column characteristics, replace width with internal diameter;

  1. Correction done.

6) 2.6 sub section, line 220. Apply the same spacing between letter, symbol and numeric value when you indicate the significance. Sometime you have written p

  1. Correction done.

Regards

Round 2

Reviewer 1 Report

Revise the manuscript. spread ratio missing?

Author Response

Thank you for your comment

Reviewer 2 Report

now it is better than before.

Author Response

Thank you for your comment.

Your suggestions were included in the text.

Reviewer 3 Report

  1. Equation for calculation whiteness index is not correct.
  2. Discussion on TPC, DPPH and ABTS is improved, although it is not the best. 

Author Response

Reviewer 3

Thank you for your comments.

1.Equation for calculation whiteness index is not correct.

W = 100-[(100-L)2 + (a2 + b2)]1/2 where L, a, and b refer to coordinates in Hunter's L, a, b Color Difference Equation.

DOI: https://doi.org/10.1007/978-1-4419-6247-8_12823

The Equation was modified in material and methods section. We wrote the equation wrong, but the results are correct.

2.Discussion on TPC, DPPH and ABTS is improved, although it is not the best.

The discussion on TPC, DPPH and ABTS was improved.